

# Estimation of ship emission rates at a major shipping lane by long path DOAS measurements

Kai Krause[1], Folkard Wittrock[1], Andreas Richter[1], Stefan Schmitt[3, 4], Denis Pöhler[3, 4], Andreas Weigelt[2], and John P. Burrows[1]

[1]Institute of Environmental Physics, University of Bremen, Germany
[2]Federal Maritime and Hydrographic Agency (BSH), Hamburg, Germany
[3]Institute of Environmental Physics, University of Heidelberg, Germany
[4]Airyx GmbH, Heidelberg, Germany

**Correspondence:** Kai Krause (kakrau@iup.physik.uni-bremen.de)

**Abstract.** Ships are an important source of $SO_2$ and $NO_x$, which are key parameters of air quality. Monitoring of ship emissions is usually carried out using in situ instruments on land, which depend on favourable wind conditions to transport the emitted substances to the measurement site. Remote sensing techniques such as long path DOAS (LP-DOAS) measurements can supplement those measurements, especially in unfavourable meteorological conditions. In this study one year of LP-DOAS

measurements made across the river Elbe close to Hamburg (Germany) have been evaluated. Peaks (i.e. elevated concentrations) in the $NO_2$ and $SO_2$ time series were assigned to passing ships and a method to derive emission rates of $SO_2$, $NO_2$ and $NO_x$ from those measurements using a Gaussian plume model is presented. 7402 individual ship passages have been monitored and their respective $NO_x$, $SO_2$ and $NO_2$ emission rates have been derived. The emission rates, coupled with the knowledge of the ship type, ship size and ship speed have been analysed. Emission rates are compared to emission factors from previous

studies and show good agreement. In contrast to emission factors (in gram per kilogram fuel) the derived emission rates (in gram per second) do not need further knowledge about the fuel consumption of the ship. To our knowledge this is the first time emission rates of air pollutants from individual ships have been derived from LP-DOAS measurements.

## 1 Introduction

Shipping plays an important role in the transport of goods around the world, with 80 - 90 % of world trade being carried by

ships. Although shipping is an efficient way of transport, the total number of ships and the relatively high emission factors of air pollutants of ship engines, have an impact on the environment and human health (Alföldy et al., 2013). The contribution of ship emissions to the global emissions of $NO_x$ and $SO_2$ was estimated to be about 15 % and 4-9 %, respectively (Eyring et al., 2010). $NO_x$ emissions are high because of the design of the engines, which operate at high temperatures and pressures (Balzani Lööv et al., 2014). $SO_2$ emissions are high because of the high fuel sulphur content of the typically used shipping

fuel (Balzani Lööv et al., 2014). $NO_x$ and $SO_2$ emissions are nowadays limited by the International Maritime Organization (IMO) Marpol, Annex VI protocol, which sets global limits for fuel sulphur content and $NO_x$ engine power-weighted emission rate. Further-more, emission control areas (ECAs) have been established in some regions, enforcing more strict emission rules.





For example the Baltic Sea, the North Sea, the English Channel and the coasts of the US and Canada are designated as ECA (emission control area) (Beecken et al., 2014). Most of the emissions caused by international shipping take place within 400 km of land and therefore have an impact on coastal air quality (Eyring, 2005). Due to the importance of ship emissions, a large number of studies has been performed previously. Measurements of air pollution, and consequently shipping emissions, are often performed with in situ instruments (e.g. Moldanová et al. (2009), Alföldy et al. (2013), Diesch et al. (2013), Beecken et al. (2014), Pirjola et al. (2014), Beecken et al. (2015), Kattner et al. (2015), Kattner (2019)), but remote sensing techniques such as differential optical absorption spectroscopy (DOAS) have also been successfully applied (e.g. Berg et al. (2012), Seyler et al. (2017), Seyler et al. (2019), Cheng et al. (2019)). Additionally, the impact of shipping emissions has been investigated by modelling studies (e.g. Eyring (2005), Ramacher et al. (2018), Ramacher et al. (2020), Tang et al. (2020)). In order to model the influence of ship emissions on air quality, one needs to characterize the international shipping fleet and to prescribe the emission behaviour of individual vessels. The information needed for this usually comes from in situ measurements, either on-board the ship or onshore. In both cases the statistics are limited, on-board measurements being restricted to a small number of ships, and onshore measurements depending on favourable wind conditions, to transport the emitted substances to the measurement site. Remote sensing techniques such as long path DOAS (LP-)DOAS can help to supplement in situ measurements, as the technique enables ship plumes, containing pollutants to be measured independent of meteorological conditions.

In this study, an approach to determine absolute emission rates of $NO_x$, $NO_2$ and $SO_2$ from LP-DOAS measurements is presented. The derived emission rates provide insight into the emission behaviour of the ship fleet entering the harbour of Hamburg, Germany, which is one of the largest ports in Europe.

## 2 Measurements and methods

### 2.1 Measurement site

Measurements made in this study were carried out in Wedel, a small town close to Hamburg, which is located on the river banks of the river Elbe. The river serves as the entrance route to the port of Hamburg and is well frequented by different types of ships going from or to Hamburg through the North Sea or the Kiel Canal. Most ships are container vessels, tankers, bulk carriers or reefer vessels. The measurement site is located on the northern banks of the river Elbe on the premises of the Waterways and Shipping Office (WSA) (53.570° N, 9.69° E) and is operated by the Federal Maritime and Hydrographic Agency (BSH) to monitor shipping emissions compliance according to MARPOL Annex VI. The standard instrumentation consists of in situ instruments to measure concentrations of $SO_2$, $CO_2$, $NO_x$ and $O_3$. Those measurements are supplemented by an AIS (automatic identification system) receiver to obtain information about the passing ships, as well as meteorological measurements. All instruments are located close to the main shipping lane with a line of sight distance to the ships steaming from or to the port of Hamburg of 300 to 500 m. The port of Hamburg is located 10 km upriver from the measurement site and the ships still or already use their main engine. The prevailing wind directions in the area, which are from the south, are such that the emissions from shipping are often blown towards the measurement site. The southern river bank is rural and sparsely



**Table 1.** Characteristics of the LP-DOAS system.

| Component | Details |
|---|---|
| Light source | Laser-Driven light source Energetiq EQ99 |
| Optical fibres | 200 μm, 800 μm |
| Telescope mirror | Diameter 0.3 m, focal length 1.5 m |
| Spectrometer | Acton Spectra Pro 300i |
| CCD | 2048×512 pixel Roper scientific back-illum. |
| Measured wavelengths | 280 – 362 nm, 0.53 nm resolution |

populated without large sources of air pollution. A detailed description of the in situ instruments used on site can be found in Kattner et al. (2015) who used these data to derive fuel sulphur content for passing ships.

## 2.2 LP-DOAS instrument

To monitor shipping emissions by optical remote sensing, a LP-DOAS instrument was set up on the northern river bank in April 2018. The instrument uses an artificial light source to emit a beam of light across the river, which is reflected by an array
of retro reflectors that is mounted onto a lighthouse on the southern river bank at an altitude of approximately 35 m above ground level. The distance between emitting telescope and retro reflector array is 2.87 km, leading to a total light path of 5.74 km. The reflected light beam is then measured and evaluated using the DOAS technique, which is explained in the next section. The technical details of the instrument are summarized in Table 1. The measurement geometry across the river is shown in Figure 1.

The instruments measurement cycle consists of one reference lamp spectrum followed by four blocks of 32 atmospheric spectra. After each block an atmospheric background spectrum is measured. Each spectrum consists of ten individual scans, which are co-added. The exposure time of these individual scans is tied to a fixed saturation of the CCD, with a maximum exposure time of 0.3 seconds. This results in a temporal resolution of 3 seconds.

## 2.3 Differential Optical Absorption Spectroscopy (DOAS)

The basic principle of spectroscopic measurements is given by Lambert-Beer's law which describes the absorption of electro-magnetic radiation by matter:

$$I(\lambda) = I_0(\lambda) \cdot exp(-\sigma(\lambda) \cdot c \cdot L), \tag{1}$$

where $I_0(\lambda)$ is the initial intensity, $I(\lambda)$ is the intensity after passing through a medium of a given thickness $L$ containing the absorbing species in concentration $c$ and $\sigma(\lambda)$ is the absorption cross-section at a given wavelength $\lambda$. In DOAS the
absorption cross-section is separated into two parts, the first one describing broadband absorption and elastic scattering can be





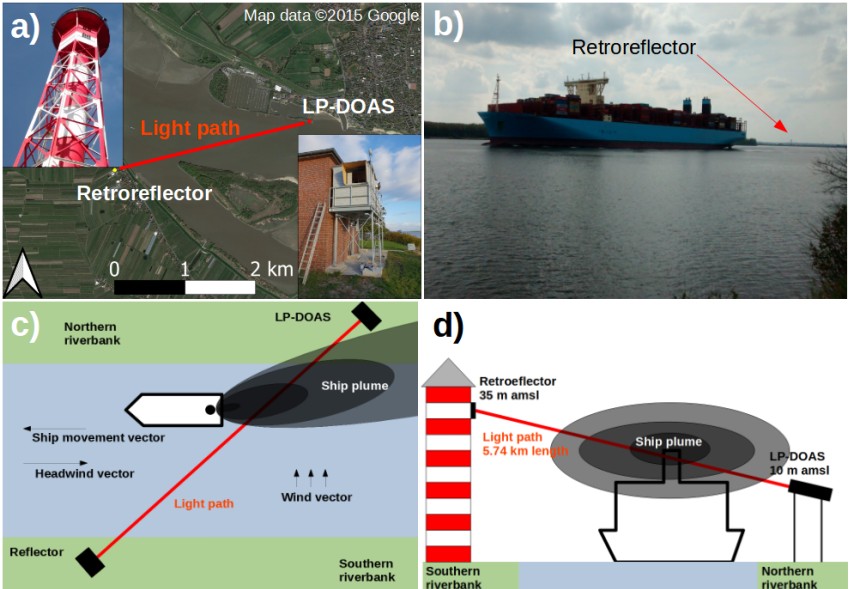

**Figure 1.** a) Satellite image of instrument location, with LP-DOAS marked as a red dot on the northern river bank and retro reflector position marked as yellow dot on the southern river bank. b) Image of a passing container ship next to the measurement site. c) Schematic overview of the measurement geometry of the LP-DOAS for a passing ship leaving Hamburg towards the North Sea, seen from above. d) same as c) but seen from the port of Hamburg. Note that c) and d) are not to scale.

approximated by a polynomial, while the second part ($\sigma'$), called differential cross-section, contains the narrow-band absorption structures. In the presence of N absorbing species, each of them has to be included with their respective absorption cross-section. Taking this into account, the DOAS equation results:

$$\ln \frac{I_0(\lambda)}{I(\lambda)} = \sum_{i=1}^{N} L \cdot c_i \cdot \sigma'_i - \sum_{p} \cdot a_p \cdot \lambda^p. \tag{2}$$

A polynomial and the differential cross sections of all relevant absorbing species are then fitted to the measured optical depth $\ln \frac{I_0(\lambda)}{I(\lambda)}$, resulting in the coefficients of the polynomial $a_p$ and the integrated number densities of the respective absorbers along the light path $L \cdot c_i$. From this quantity, the concentration can be determined because $L$ is known from the experiment set-up, $c_i$ being the mean concentration of species $i$ along $L$.

## 2.4 Data analysis

In order to derive the contribution of an individual ship to the total measured integrated concentration of a pollutant, further steps are needed. The data analysis comprises three steps. Firstly the measured spectra were analysed using the DOAS tech-





**Table 2.** DOAS fit settings for the retrieval of $SO_2$, $NO_2$ and $O_3$.

| Trace gas | $SO_2$ | $NO_2$ | $O_3$ |
|---|---|---|---|
| Fit window | 297.0 - 309.0 nm | 334.5 - 356.5 nm | 282.0 - 314.5 nm |
| Polynomial degree | 3 | 3 | 3 |
| Cross sections | $NO_2$ (Vandaele et al., 1996) $O_3$ (Serdyuchenko et al., 2014) $SO_2$ (Vandaele et al., 1996) HCHO (Meller and Moortgat, 2000) | $NO_2$ (Vandaele et al., 1996) $O_3$ (Serdyuchenko et al., 2014) $O_4$ (Thalman and Volkamer, 2013) HCHO (Meller and Moortgat, 2000) HONO (Stutz et al., 2000) | $NO_2$ (Vandaele et al., 1996) $O_3$ (Serdyuchenko et al., 2014) $SO_2$ (Vandaele et al., 1996) HCHO (Meller and Moortgat, 2000) |

nique to determine the concentration of the absorbing gas along the path of the electromagnetic radiation. Secondly the elevated concentrations which we attribute to a particular passing ship are estimated. Thirdly the emission rate of the trace gas for the assigned ship is calculated.

The fit settings to retrieve $NO_2$ and $SO_2$ time series from the measured spectra are shown in Table 2.

An example time series of the fitted trace gases is shown in Figure 2. The blue lines show the fitted trace gas time series and the orange lines the corresponding detection limit. The gray dashed lines mark passing ships that have been assigned to a peak. The green lines shows the estimated background concentration. Following Stutz and Platt (1996), the DOAS measurement error was defined as two times the DOAS fit error and the detection limit for the trace gases was defined as two times the

measurement error (four times the DOAS fit error), which results in a median detection limit of 190 pptv for $NO_2$, 59 pptv for $SO_2$ and 253 pptv for $O_3$.

### 2.4.1    Peak identification

For each emission plume, measured maxima, i.e. enhanced amounts of $NO_2$, $SO_2$ and a minimum, i.e. a diminished amount of $O_3$ values are found, as expected. To identify such peaks, a low pass filtered time series is calculated using a running median

with a windows size of five minutes. The low pass filtered time series represents the background concentration including influences by meteorological factors but excludes the short term variations caused by plumes of passing ships. The low pass filtered time series is then subtracted from the original time series, resulting in a time series which is close to zero on average, but contains several peaks. If one of those peaks exceeds a predefined threshold, then the peak is marked as a valid increase in trace gas concentration caused by some sort of emission, e.g. a passing ship. In this study the threshold was set to four times





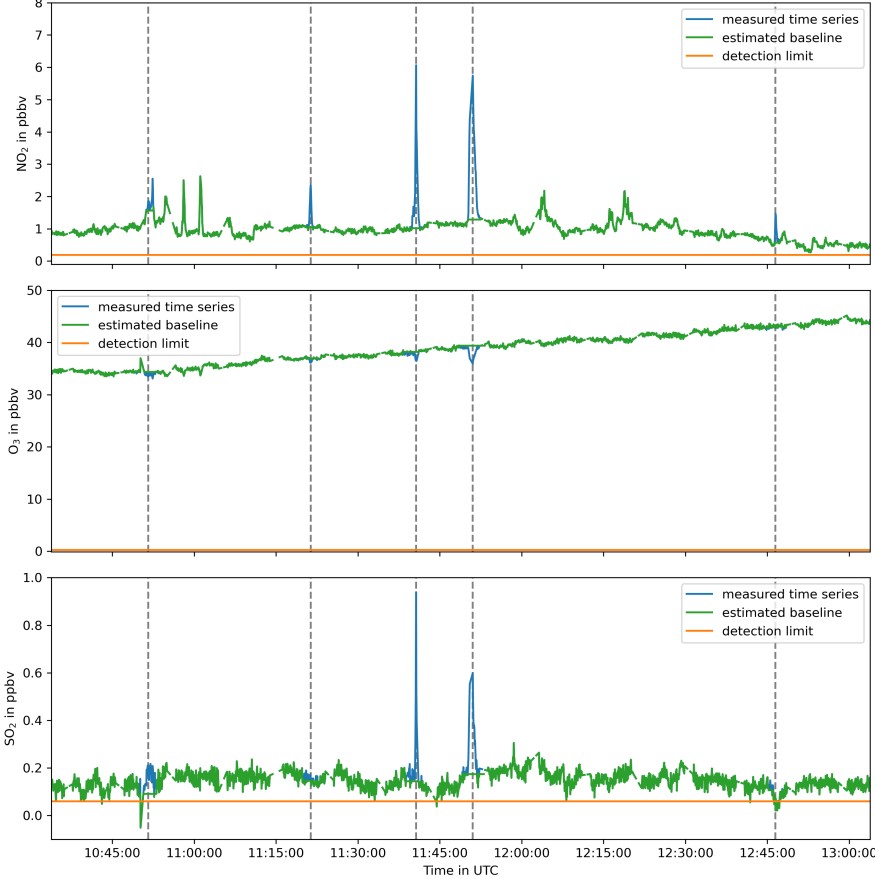

**Figure 2.** Example time series of the fitted trace gases from 19th August 2018 between 10:30 and 13:00 UTC. In each plot the blue line shows the fitted time series of the respective trace gas and the orange line shows the respective median detection limit. The grey dashed lines mark passing ships, that have been assigned to a peak in the time series. The green line shows the calculated background.

the DOAS fit error of the respective trace gas measurement. This analysis is carried out separately for each relevant trace gas ($NO_2$, $SO_2$, and $O_3$).

In the next step, the identified peaks are assigned to individual ships. The assignment is based on AIS data of passing ships. AIS data contains the current position, speed and heading of the ship, as well as other, more general information about the ship itself (e.g. speed, course, type, length, destination, ...). The AIS data is transmitted in regular intervals of two to thirty

seconds and is interpolated to one second time resolution using linear interpolation between two received AIS messages. For each detected peak in the trace gas time series it is then checked, if there was a ship in a position that could have caused the increase in trace gas concentration. If there is a single ship in a position that could be the source of the enhancement of the trace gas concentration, this ship is assigned to the respective peak. The assignment is based on position and time. For each peak occurrence $t_{peak}$, a time window of $(t_{peak} - \Delta t_{before}) < (t_{peak} + \Delta t_{after} + \Delta t_{dyn})$ is defined, where $\Delta t_{before}$ is set to 30 and





$\Delta t_{after}$ is set to 120 seconds, and $\Delta t_{dyn}$ is calculated as the length of the ship divided by the speed of the ship. The windows starts before peak occurrence to accommodate for ship plumes that are transported by wind through the light path before the ship itself passes the light path. The windows are extended dynamically by ship size and ship speed to incorporate that larger ships may need a longer time to pass through the light path. Due to the length of the defined time window, several AIS positions of an individual ship are possible source positions. The final assignment is based on distance to the light path as well as course

and length of the ship. The first position where the ship could have fully passed the light path is assigned as the ship position responsible for the trace gas peak. In median the time difference between measurement of the peak maximum and the assigned AIS signal is 20 seconds. The approach fails if the traffic density of ships is too high, making the unambiguous attribution of a plume to a particular ship impossible. Evaluation of the AIS data shows that on average there are 110 ship passages per day.

### 2.4.2    NO$_2$ to NO$_x$ conversion

Being restricted to the wavelength range between 280 and 360 nm, the LP-DOAS measures NO$_2$, while the ship emits nitrogen oxides (NO$_x$) as NO and NO$_2$. A part of the NO$_2$ is produced during the combustion process and emitted by the ship directly, while another part is formed after emission by reaction with ozone in the atmosphere:

$$NO + O_3 \rightarrow NO_2 + O_2. \tag{R1}$$

     To estimate the total NO$_x$ emission, a simple approach is used to convert the measured NO$_2$ concentrations to NO$_x$ con-

centrations. The correct NO$_2$/NO$_x$ ratio can be obtained by summing the NO$_2$ and O$_3$ signals and plotting this sum against the measured NO$_x$ concentration (Clapp, 2001; Kurtenbach et al., 2016). This kind of analysis has been carried out using data from the in situ measurements which provide NO$_x$, NO$_2$ and O$_3$ observations and results in a mean NO$_2$/NO$_x$ ratio of 0.138 (see Figure 3) which agrees with previous studies Cooper (2001). This means most of the emitted NO$_x$ is emitted as NO and only a smaller fraction is directly emitted as NO$_2$. The NO$_2$ peak observed by LP-DOAS can then be converted to NO$_x$ using

the following formula:

$$NO_x = \frac{(\Delta NO_2 + \Delta O_3)}{NO_2/NO_x ratio} \tag{3}$$

     where $\Delta NO_2$ is the increase in NO$_2$ caused by the ship and $\Delta O_3$ is the decrease in O$_3$ caused by the reaction of emitted NO with atmospheric O$_3$ and is also measured by the LP-DOAS. Using this approach, the total amount of measured NO$_2$ is corrected for the NO$_2$ that formed during transport in the atmosphere, and the remaining NO$_2$ is the amount primarily emitted

by the ship. This primarily emitted NO$_2$ is then used to estimate the amount of emitted NO$_x$ using again the NO$_2$/NO$_x$ ratio. This procedure assumes that the NO$_2$/NO$_x$ ratio is the same for all ships and that no other species are emitted which could impact on the NO$_2$ production or ozone removal. Based on the compact correlation found (see Figure 3), these assumptions appear to be justified.



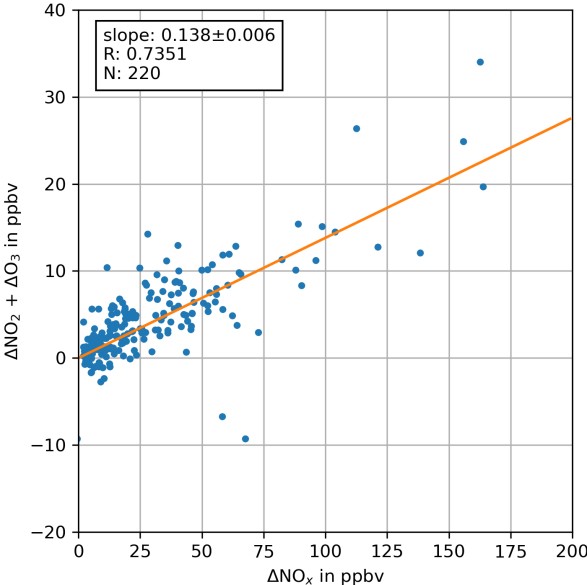

**Figure 3.** Plot of $\Delta NO_2 + \Delta O_3$ against $\Delta NO_x$ from peaks measured with the in situ instruments between April 2018 and May 2019. All concentrations have been corrected for background concentrations. For this analysis, 194 manually quality checked peaks were used. This results in a slope (a $NO_2/NO_x$ ratio) of 0.138 with a respective standard error of 0.006.

### 2.4.3 Estimation of emission rate

As the LP-DOAS instrument does not measure the concentration of the trace gases at the stack, a model has to be applied to estimate the emission from the concentration enhancement found for a given light path. This conversion is based on the assumption that the plume of a single ship can be described by a simple Gaussian plume model (Pasquill, 1968) and can be expressed mathematically by Eq. 4:

$$C(x,y,z) = \frac{Q}{2\pi U \sigma_y \sigma_z} \cdot \exp\left(\frac{-y^2}{2\sigma_y^2}\right) \cdot \left[exp\left(\frac{-(z-H)^2}{2\sigma_z^2}\right) + exp\left(\frac{-(z+H)^2}{2\sigma_z^2}\right)\right] \tag{4}$$

where $Q$ is the emission rate of a substance in gram per second, $U$ is the wind speed in meter per second along the main wind direction (aligned with x), $\sigma_y$ and $\sigma_z$ are the dispersion parameters in horizontal (y) and vertical (z) direction in meter and $H$ is the height of the plume center in meter. The dispersion parameters depend on $x$, the atmospheric stability and the surrounding environment, which differs for open country and urban conditions. A simple classification scheme for the stability classes is shown in Table 3, while the corresponding dispersion parameters are listed in Table 4.

To determine atmospheric stability at the measurement site the wind speed measurements of the in situ instruments are used, while incoming global radiation and cloud coverage are taken from a nearby measurement station of the German Weather Service located at the Hamburg-Airport (DWD Climate Data Center (a), DWD Climate Data Center (b)).




**Table 3.** Atmospheric stability classification scheme based on surface wind speed and solar insulation for day time conditions and cloud cover during night time conditions (Pasquill, 1968). Ranging from very unstable (A) to moderately stable (E).

| Surface wind speed 10 m a.g.l. ($\mathrm{m\,s^{-1}}$) | Daytime solar radiation | | | Night time cloud cover | |
|---|---|---|---|---|---|
| | Strong | Moderate | Slight | >= 4/8 clouds | <= 3/8 clouds |
| < 2 | A | A - B | B | - | - |
| 2-3 | A - B | B | C | E | F |
| 3-4 | B | B - C | C | D | E |
| 4-6 | C | C - D | D | D | D |
| > 6 | C | D | D | D | D |

**Table 4.** Atmospheric dispersion parameters $\sigma_y$ and $\sigma_z$ for different stability classes in dependence of distance (x) from source in meter. For intermediate cases such as A - B the average of both values has been taken (Briggs, 1973).

| Stability class | $\sigma_y(x)$ | $\sigma_z(x)$ |
|---|---|---|
| A | $0.22x(1+0.0001x)^{-0.5}$ | $0.20x$ |
| B | $0.16x(1+0.0001x)^{-0.5}$ | $0.12x$ |
| C | $0.11x(1+0.0001x)^{-0.5}$ | $0.08x(1+0.0002x)^{-0.5}$ |
| D | $0.08x(1+0.0001x)^{-0.5}$ | $0.06x(1+0.0015x)^{-0.5}$ |
| E | $0.06x(1+0.0001x)^{-0.5}$ | $0.03x(1+0.0003x)^{-0.5}$ |
| F | $0.04x(1+0.0001x)^{-0.5}$ | $0.016x(1+0.0003x)^{-0.5}$ |

In order to calculate the emission rate of a ship, the model is evaluated once with an arbitrary emission rate ($Q_{model}$), using the ship's position as the starting point of the plume. The effective height of the plume center is set to the height of the funnel above water level assuming that the plume quickly bends down due to wind and the movement of the ship. As the height of the ship stack is not transmitted in the AIS, the height of the stack is estimated from pictures of the respective ship, preferably taken by the camera of one of the instruments, otherwise pictures from marinetraffic.com were used (MarineTraffic). Dispersion parameters $\sigma_y$, $\sigma_z$ are chosen according to atmospheric stability. To account for the movement of the ship, the wind speed and direction have been combined with the ship movement to an apparent wind speed and apparent wind direction (Berg et al., 2012):

$$U_{aw} = \sqrt{(v_{wind\,N} + v_{ship\,N})^2 + (v_{wind\,E} + v_{ship\,E})^2} \tag{5}$$

$$\theta_{aw} = -atan2[(v_{wind\,E} + v_{ship\,E}), (v_{wind\,N} + v_{ship\,N})] \tag{6}$$



where $v_{wind\,E}$, $v_{ship\,E}$ and $v_{wind\,N}$, $v_{ship\,N}$ are the eastern and northern velocity components of the wind vector and ship movement vector, respectively.

As the real emission rate ($Q_{meas}$) is unknown, this model run only gives insight into the dispersion of the emitted species. In order to retrieve the desired emission rate for a certain species emitted by the ship, the concentration measured at the measurement site ($C_{meas}$) is compared to the modelled concentration ($C_{model}$) along the light path, where $C_{meas}$ is the already background corrected measured trace gas concentration. The correction is applied to each peak individually and is carried out by subtracting the mean concentration 30 seconds before and after the peak from the peak itself. The low-pass filtered time series used to identify the peaks is not used as a background, as it may overestimate the background concentration in cases of high traffic density. As the LP-DOAS measurement is integrating along the light path $C_{model}$ is obtained by averaging all model grid cells along a path through the model grid, which corresponds to the light path during the measurement (see Figure 1). Assuming all parameters are estimated correctly, the only difference between modelled concentration and measured concentration is caused by a different emission rate. Therefore $Q_{meas}$ can be estimated by the following equation:

$$Q_{meas} = \frac{C_{meas}}{C_{model}} \cdot Q_{model} \tag{7}$$

This approach assumes that the motion vector of the ship and the emission rate is constant for the time between emission and measurement of the enhanced concentration.

### 2.4.4 Estimation of uncertainty

The uncertainty of the emission rate is given by:

$$\sigma_Q = \sqrt{\left(\frac{\partial Q_{meas}}{\partial C_{meas}} \cdot \sigma_{C\,meas}\right)^2 + \left(\frac{\partial Q_{meas}}{\partial C_{model}} \cdot \sigma_{C\,model}\right)^2} \tag{8}$$

where $\sigma_{C\,meas}$ is the uncertainty of the measured trace gas concentration and $\sigma_{C\,model}$ the uncertainty of the modelled trace gas concentration. In case of $NO_x$, $\sigma_{C\,meas}$ consists of the uncertainty of the $NO_2$ concentration, the uncertainty of $O_3$ and the uncertainty of the $NO_2/NO_x$ ratio.

To calculate $\sigma_{C\,model}$, Monte-Carlo-Simulations are performed for each individual passing ship, where $U_{aw}$, $\theta_{aw}$, atmospheric stability, latitudinal- and longitudinal position of the ship and the funnel height of the ship are varied within their respective uncertainty range. The assumed uncertainty for each parameter is shown in Table 5. This results in a set of simulations for every input parameter, and for each simulation in the respective set, the concentration along the artificial light path is determined. A set for a single input parameter ($j$) is then summarized as mean concentration ($mean_{C\,j}$), the respective standard deviation ($\sigma_{C\,j}$), minimum ($min_{C\,j}$) and maximum value ($max_{C\,j}$). The model uncertainty is then calculated as:

$$\sigma_{C\,model} = \sqrt{\sigma_{C\,U\,aw}^2 + \sigma_{C\,\theta\,aw}^2 + \sigma_{C\,stability}^2 + \sigma_{C\,lon}^2 + \sigma_{C\,lat}^2 + \sigma_{C\,H}^2} \tag{9}$$





where each $\sigma_{Cj}$ is the standard deviation of the modelled trace gas concentrations of the Monte-Carlo-Simulations with respect to changes of an individual parameter $j$. As the parameters are changed individually, possible interactions between changes of more than one parameter at a time are neglected.

The largest error source is the uncertainty of the position of the emission source, as it has a large impact on which part of the plume is assumed to be measured, and thus has a large impact on the derived $C_{model}$. The position is determined by the data transmitted by the AIS with an average error of 10 m or less. However, since the the location of the funnel in relation to the AIS transmitter on the ship itself is not known, the positional error also depends on the dimensions and orientation of the ship. For the calculation it is assumed that the emission source is located at the position given by the AIS. It is assumed that the transmitter is located at or close to the bridge of the ship, and that the funnel is also close to it. For smaller ships this is certainly true, due to the small dimensions of the ship. For larger ships such as tankers and container ships, different designs exist. It is assumed that the transmitter is close to the bridge here as well, and that the main exhaust is not further away from that position than half the ship width or length.

Additionally the height of the emission depends on the stack height and the water level. The stack height is estimated from pictures of the ship which gives an initial uncertainty of the value, further more the height above water level depends on the draft of the ship, which is transmitted by the AIS. At the measurement site the water level of the river Elbe is not only influenced by the amount of water flowing downstream but also by the tide. The water level is assumed to be between the long-term mean high and mean long-term low water level.

The second largest source of errors is the apparent wind used in the calculation. The apparent wind itself is calculated from the horizontal wind velocity vector and the ship velocity vector. In most cases, the magnitude of the ship velocity vector is large compared to the wind velocity vector, and therefore the uncertainty is dominated by the uncertainty of the ship's speed and course. The smallest error source is the uncertainty of the derived trace gas time series.

For $NO_x$, the uncertainty of the derived $NO_2/NO_x$ ratio is another important factor for the overall uncertainty of the derived emission rate. As the $NO_2/NO_x$ ratio is in the denominator of equation 3, even a small uncertainty of the ratio can lead to significant changes in the estimated $NO_x$ concentration. Generally it is assumed the $NO_2/NO_x$ ratio is the same for all ships, which is also supported by the compact correlation found in Figure 3, but the ship type and operation mode of the engine can also have an influence on this ratio.

## 3  Results

Between April 2018 and May 2019 a total number of 7402 passing ships were identified and assigned to a peak in the trace gas time series. Due to technical problems with the instrument, there were only 233 days of measurements during this time period. Most of the measurements took place between June 2018 and February 2019, while before and after there were only individual days of measurements. For each ship passage, emission rates of $SO_2$, $NO_2$ and $NO_x$ were calculated. This dataset has then been filtered to remove non-physical results such as very high emission rates of several tons per second. These non-physical





**Table 5.** Uncertainties of the input parameters used in the Monte-Carlo-Simulations.

| Abbreviation | Name | Calculation of value |
|---|---|---|
| $\sigma_{lon}$ | ship extent in longitudinal direction | $\frac{1}{2} \cdot (|length \cdot \sin(heading)| + |width \cdot \cos(heading)|)$ |
| $\sigma_{lat}$ | ship extent in latitudinal direction | $\frac{1}{2} \cdot (|length \cdot \cos(heading)| + |width \cdot \sin(heading)|)$ |
| $\sigma_H$ | plume height | $\sqrt{\sigma_{fh}^2 + \sigma_{wl}^2}$ |
| $\sigma_{fh}$ | funnel height | estimated: 5 m |
| $\sigma_{wl}$ | water level | mean high water level - mean low water level |
| $\sigma_{aw}$ | apparent wind speed | $\sqrt{\sigma_{v_{windN}}^2 + \sigma_{v_{shipN}}^2 + \sigma_{v_{windE}}^2 + \sigma_{v_{shipE}}^2}$ |
| $\sigma_{v_{windN}}$ | wind speed | standard deviation of northern wind component |
| $\sigma_{v_{windE}}$ | wind speed | standard deviation of eastern wind component |
| $\sigma_{v_{shipN}}$ | ship speed | estimation based on $0.514\,\mathrm{m\,s^{-1}}$ uncertainty in speed |
| $\sigma_{v_{shipE}}$ | ship speed | and 10 ° uncertainty in heading |
| $\sigma_{\theta_{aw}}$ | apparent wind direction | estimated: 10 ° |
| $\sigma_{stability}$ | stability | atmospheric dispersion parameters of class with lower stability and higher stability than the assigned class |
| $\sigma_{NO_2/NO_x}$ | NO$_2$/NO$_x$ ratio | standard error of the slope (0.006) |
| $\sigma_{c_{meas}}$ | DOAS measurement error | individual DOAS measurement error for each trace gas |
|  |  | mean value for NO$_2$ 1.5 % |
|  |  | mean value for SO$_2$ 17.7 % |
|  |  | mean value for O$_3$ < 0.1 % |

values occur when the assumptions within the Gaussian plume model do not reflect the situation during the measurement and therefore the shape of the calculated plume does not match the real plume.

To eliminate such cases before further investigation, three criteria have been defined. If one of these criteria is violated for a single input parameter $j$ for a given individual measurement, the derived emission rate is omitted from the further analysis. The criteria are:

1) $mean_{Cj}/C_{model}$ has to be between 0.8 and 1.2, to eliminate cases where the uncertainty introduced by the input parameter systematically leads to too high or low derived concentrations.

2) $\sigma_{Cj}/C_{model}$ has to be lower than 0.4, to eliminate cases that have a high variability if input parameters are varied within their uncertainties.

3) The difference between $max_{Cj}/C_{model}$ and $min_{Cj}/C_{model}$ has to be smaller than 1, to eliminate cases with a large spread between minimum and maximum value.

After this quality check a total of 886 NO$_x$, 1069 SO$_2$ and 1375 NO$_2$ emission rates were left for further analysis, and the
emission rates have an uncertainty of 43 % in the mean and 35 % in median. The exclusion of many measured ships is a result of the here applied plume model and not due to the LP-DOAS measurement itself. It is part of the current developments to

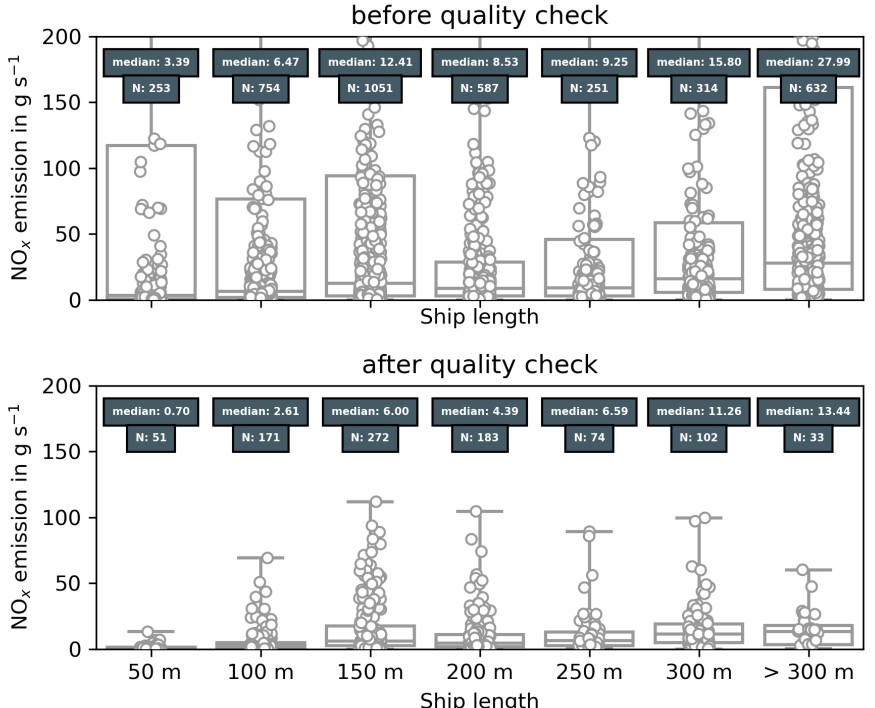

**Figure 4.** Box plot of $NO_x$ emission rates in $g\,s^{-1}$ for different ship sizes. Boxes indicate the 25% and 75% percentile, the line in the middle is the median and the bars show minimum and maximum values. Dots show individual measurements. Dark grey boxes show the median emission rate and total number of observations for this length class.

increase the output rate with different measurement configurations. The total number of ships differs because the assignment of a ship to a peak in the trace gas time series is carried out for each trace gas individually, which leads to some differences between $SO_2$ and $NO_2$ signal strength. The sulphur content of shipping fuel is limited to $0.10\,\%\,S\,M/M$ for seagoing and

$1\times10^{-5}\,\%\,S\,M/M$ for inland ships, resulting in ship passes which clearly cause a peak in $NO_2$ while the enhancement in $SO_2$ is too low to be detected as a peak. For $NO_x$ the concentrations of $\Delta NO_2$ and $\Delta O_3$ are summed and under circumstances with a high temporal variability within those trace gas concentrations, the background correction for the individual peaks might be erroneous and thus the sum can be zero or even negative. In such cases the $NO_x$ emission rate is not calculated. As an example, Figure 4 shows the difference between the unfiltered and filtered dataset for $NO_x$ emission rates for different ship length classes

as box plots.

     The unfiltered data set shows a large variability, indicated by the large boxes, while the filtered data set clearly shows a lower variability and a narrow distribution around the median of the respective length class. An exception is the 150 m length class, which still shows a high variability. This variability is caused by dredging ships and will be discussed in more detail in one of the next paragraphs.





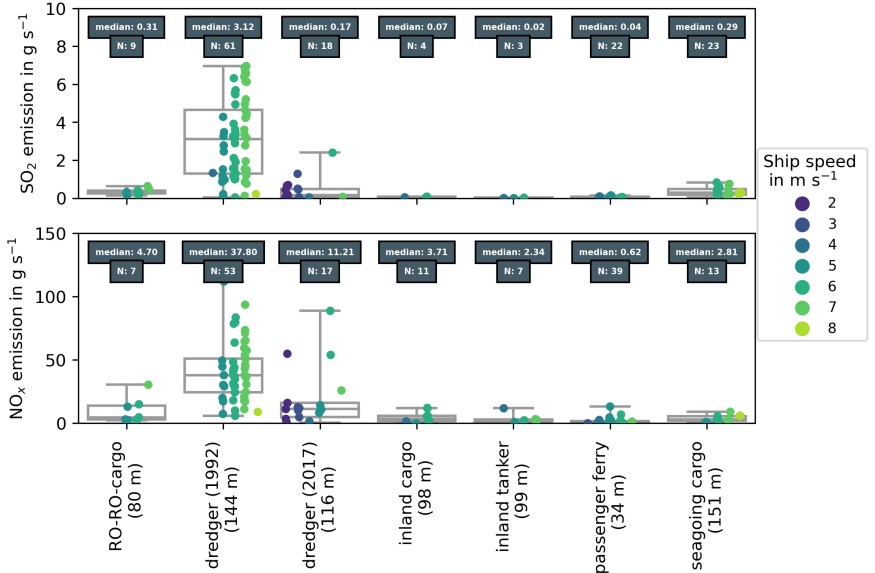

**Figure 5.** Box plot of $SO_2$ and $NO_x$ emission rates in $g\,s^{-1}$ for individual ships, their respective length is given in brackets. Boxes indicate the 25% and 75% percentile, the line in the middle is the median and the bars show minimum and maximum values. Dots show individual measurements and are colour coded by corresponding ship speed. Dark grey boxes show the median emission rate and total number of observations for this ship.

There are several ships which passed the measurement site multiple times or even on a regular basis. This allows to determine the emission rate for a single ship under different measurement conditions. Emission rates of $SO_2$ and $NO_x$ for a variety of ships are shown in Figure 5 as box plots. Generally the 25% and 75% percentiles are close to their respective median values, which indicates that the estimation method works consistently. A larger variability of the emission rate for an individual ship usually indicates special operating conditions of the ship. Examples for this are the two dredging ships. These ships can operate

under varying conditions and do not necessarily only pass by, but sometimes excavate material from the bottom of the river. This might lead to higher engine loads in general or the usage of additional auxiliary engines, which in turn increases the total emission of those ships. At other times, those ships just steam through the light path without carrying out additional work, which explains the low emission rates observed on some passes. A combination of these different operating conditions leads to the high spread seen in the box plots.

Differences in emission rates between ship types can be seen. Figure 6 shows box plots of the $SO_2$ and $NO_x$ emission rates for inland ships, sea going ships and dredging ships. Generally sea going ships tend to have higher emission rates, with a median of $5.23 \pm 14.0\,g\,s^{-1}$ for $NO_x$ and $0.28 \pm 0.87\,g\,s^{-1}$ for $SO_2$, while for inland ships the median is $1.93 \pm 8.17\,g\,s^{-1}$ for $NO_x$ and $0.06 \pm 0.19\,g\,s^{-1}$ for $SO_2$. The difference in $SO_2$ can be attributed to two different factors. First of all, inland ships use fuel having a lower fuel sulphur content, which automatically decreases the amount of $SO_2$ emitted per amount of





fuel. Secondly, inland ships are smaller and have smaller engines, consuming less fuel per unit time. In combination these two factors explain the lower $SO_2$ emission rates found for inland ships. Most of the $NO_x$ formed during combustion consists of atmospheric nitrogen and oxygen. The amount of $NO_x$ formed is temperature dependent, higher engine temperatures leading to higher amounts of $NO_x$ (Alföldy et al., 2013). For inland ships there is already a limit for their $NO_x$ emissions, while for seagoing ships there was none at the time the measurements took place, which explains the higher $NO_x$ emission rates for

seagoing ships. This can also be seen in Figure 7, where the $NO_x$ emission rate is categorized for different ship size classes and the median emission rate increases with size. The emission rates are also correlated to ship speed, faster ships generally having a higher emission rate (see Figure 8). The decrease in the $SO_2$ emission rate for ship speeds larger than $7\,\mathrm{m\,s^{-1}}$ is probably caused by the low number of observations, which only include a single individual ship.

The determined median $SO_2$ emission rate for inland ships is larger than the expected $SO_2$ emissions by those ships. A

simple calculation of the expected $SO_2$ emission rate can be made by multiplying the fuel sulphur content with the amount of fuel used per unit of time. Table 7 shows those calculations for inland diesel fuel and fuel which qualifies for the SECA limit of $0.10\,\%\,\mathrm{S\,M/M}$. The observed median $SO_2$ emission rate for inland ships is $0.06\,\mathrm{g\,s^{-1}}$, which is considerably higher than the expected $SO_2$ emission rate ($0.0009\,\mathrm{g\,s^{-1}}$) for the typical fuel consumption of an inland ship and still too high when assuming the typical fuel consumption of a much larger ship.

But it has however to be kept in mind that the $SO_2$ emission rates, especially for inland ships, are biased towards high emitters, as some ships can only be identified in the $NO_2$ time series, while there is no detectable peak in the $SO_2$ time series. Ships with low $SO_2$ emissions are therefore under represented in the data set. In order to calculate a more representative mean $SO_2$ emission rate for inland ships, the total number of observed inland ships has to be taken from the $NO_2$ dataset instead, and all cases without an associated $SO_2$ emission rate are treated as zero $SO_2$ emission. The total number of observed inland

ships would then be 296 (identified from the $NO_2$ peaks and with valid $NO_2$ emission rate) and 220 of them would be treated as zero $SO_2$ emitters. This results in a mean $SO_2$ emission rate of $0.03\,\mathrm{g\,s^{-1}}$ with and median emission rate near zero, which means the $SO_2$ emissions for inland ships are often below the detection limit of the LP-DOAS instrument. For sea going ships the method works better and the median $SO_2$ emission rate ($0.28\,\mathrm{g\,s^{-1}}$) lies in the range estimated in Table 7.

For 26 individual ship passages (excluding dredging ships), the derived $SO_2$ emission rates are above the upper limit es-

timated in Table 7, which possibly indicates that those ships use fuel which does not comply with the SECA limit of $0.10$ $\%\,\mathrm{M/M}$.

Most studies derive emission factors, which specify the mass of air pollutant released per mass of burnt fuel, whereas emission rates are less commonly reported. To compare these two physical quantities, further knowledge of fuel consumption is required. Table 6 shows the derived emission rates in comparison to the results of other studies under the assumption of two

different fuel consumption scenarios. The lower value describes typical fuel consumption of an inland ship (about $165\,\mathrm{kg\,h^{-1}}$) and the upper value describes the fuel consumption of a large container ship, with carrying capacity of roughly 14.000 TEU, at a speed of $7\,\mathrm{m\,s^{-1}}$ ($2000\,\mathrm{kg\,h^{-1}}$) (Notteboom and Vernimmen, 2009), which is slightly faster than the typical speed for the largest passing ships ($6\,\mathrm{m\,s^{-1}}$).

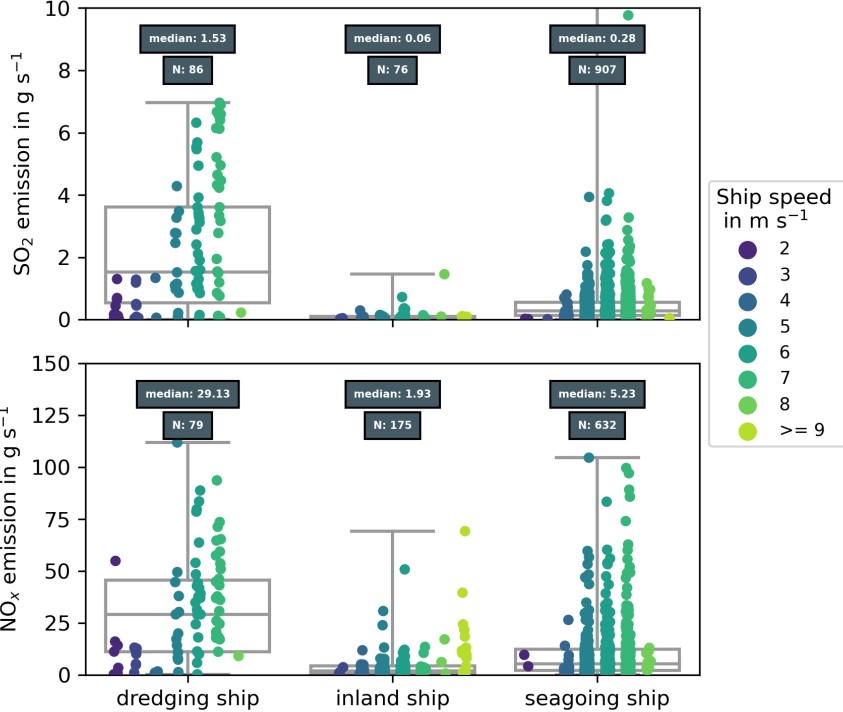

**Figure 6.** Box plot of $SO_2$ and $NO_x$ emission rates in $g\,s^{-1}$ for different ship types. Boxes indicate the 25% and 75% percentile, the line in the middle is the median and the bars show minimum and maximum values. Dots show individual measurements and are colour coded by corresponding ship speed. Dark grey boxes show the median emission rate and total number of observations for each ship type.

In all cases the median emission rate derived by our method lies within the range estimated using the emission factors of other studies, although closer to the lower bound. This is reasonable, because most passing ships are seagoing ships, with higher fuel consumption than inland ships and at the same time do not belong to the largest ship class with the highest fuel consumption.

Comparison to emission rates of Berg et al. (2012) shows larger differences. Berg et al. (2012) found a mean emission rate of $11.4 \pm 7.8\,g\,s^{-1}$ for $NO_2$ and $14.6 \pm 9.1\,g\,s^{-1}$ for $SO_2$, while in this study the mean $NO_2$ emission rate is $1.5 \pm 2.9\,g\,s^{-1}$ and the mean $SO_2$ emission rate is $0.6 \pm 1.1\,g\,s^{-1}$. This can be explained by different reasons. First of all Berg et al. (2012) observed transects of ship plume on the open seas, where the fuel sulphur limit at the time was $1.0\,\%\,M/M$, which is a factor of 10 higher than at the time of the measurements in this study, thus the emission rates of $SO_2$ should also be higher by roughly a factor of ten. Additionally ships on the open seas travel at higher speeds than at our measurement site, which increases their fuel consumption and thus their $SO_2$ emission rates. Considering the different fuel sulphur content and different speeds, both mean emission rates agree within their respective uncertainties.





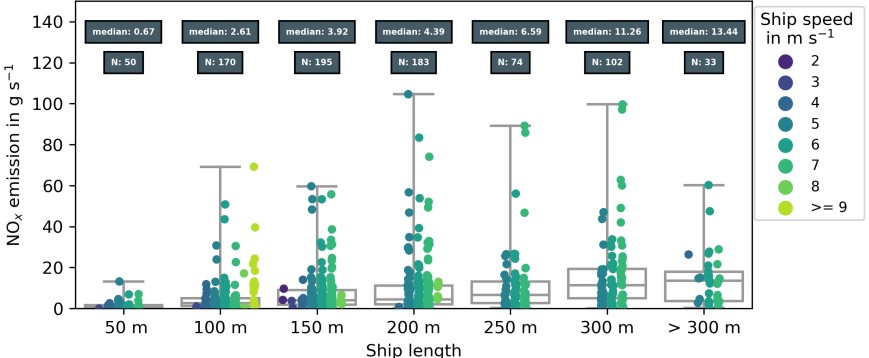

**Figure 7.** Box plot of $NO_x$ emission rates in $g\,s^{-1}$ for different ship lengths. Boxes indicate the 25% and 75% percentile, the line in the middle is the median and the bars show minimum and maximum values. Dots show individual measurements and are colour coded to corresponding ship speed. Dark grey boxes show the median emission rate and total number of observations for this length class. Data of dredging ships has been excluded.

The difference in the $NO_2$ emission rates might be also caused by the age of the observed plume, because in older plumes emitted NO can already react with atmospheric $O_3$ to form $NO_2$. In this study the plumes are measured shortly after their emission, while Berg et al. (2012) probably measured older plumes. Comparing the mean $NO_x$ emission rate ($11.0 \pm 16.1$ $g\,s^{-1}$) with the $NO_2$ emission rate of Berg et al. (2012) shows much better agreement between both. This is also supported by
our calculated $NO_2/NO_x$ ratio of 0.138, which indicates most $NO_x$ is emitted as NO which then reacts with atmospheric ozone to form $NO_2$.

In general the result for a single measurement is prone to errors. The main reasons are due to the measurement geometry and the assumptions made in modelling the plume expansion. Only a small proportion of the light path is affected by the plumes of passing ships and as the LP-DOAS measures the integrated concentration along the light path, the measurement is influenced
by the background variability along the light path. With a shorter light path, which only covers the main shipping lane and less background air masses, the enhancement in $SO_2$ and $NO_2$ would be more pronounced. This would increase the chances of detecting the plume of a passing ship, even for ships with low emission rates.

The main source of uncertainty is the plume modelling due to the uncertainty of the exact stack position and height and the simplification of turbulent structures within the plume. While the position of the ship's AIS receiver is known, the exact
position of the stack on the ship is unknown, which results in an uncertainty of the plume position and thus the modelled concentrations. A better knowledge of the exact position of the emission source would therefore increase the quality of the derived emission rates and reduce the number of omitted emission rates.

However, repeated measurements of the same ship show a low variability in the derived emission rates with the exception of the dredging ships (e.g. Figure 5). The value of the calculated emission rates lies in the large number of measured ships and



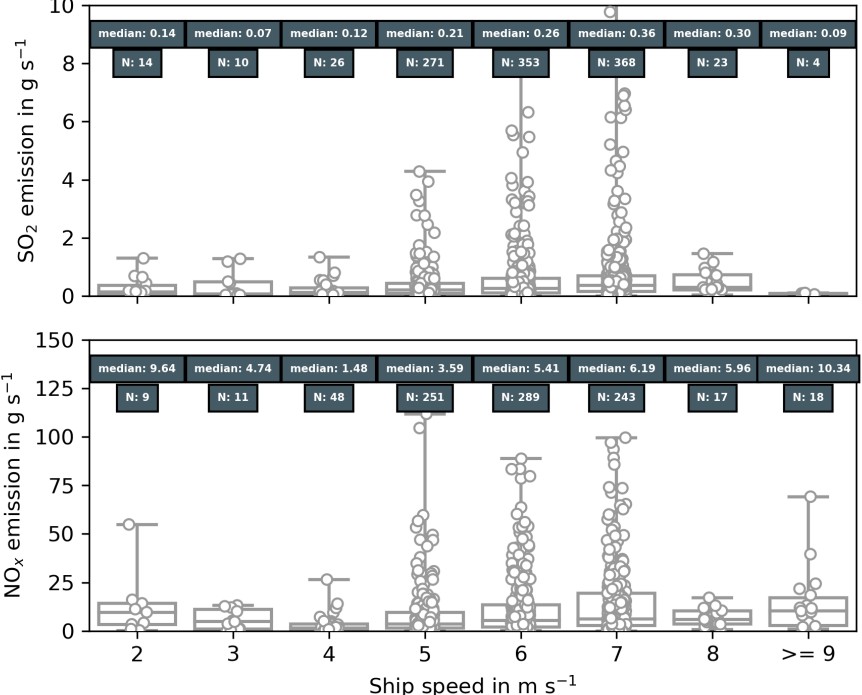

**Figure 8.** Box plot of $SO_2$ and $NO_x$ emission rates in $g\,s^{-1}$ for different ship speeds. Boxes indicate the 25% and 75% percentile, the line in the middle is the median and the bars show minimum and maximum values. Dots show individual measurements. Boxes show the median emission rate and total number of observations for each ship speed.

their statistics, which covers different meteorological conditions, and allows to characterize the emission behaviour of a fleet of ships entering the port of Hamburg.

## 4   Summary and Conclusions

A LP-DOAS instrument has been set up to measure ship emissions of $SO_2$ and $NO_2$ across the river Elbe, about 10 km seawards of Hamburg harbour. Between April 2018 and May 2019 a total number of 7402 passing ships have been identified
and assigned to peaks in the trace gas time series. A method to derive ship emission rates of different trace gases was developed and successfully applied to the measurements. The method uses a Gaussian plume model to simulate the plumes of passing ships and to derive the concentration the instrument would have measured given the assumptions made in the model. The calculated concentration is compared to the measured enhancement in the trace gas to calculate the emission rate. The derived emission rates have then to be filtered for non-physical results, which occur when the assumptions made for the model do not
reflect the measurement situation. After filtering a total, of 886 $NO_x$, 1069 $SO_2$ and 1375 $NO_2$ emission rates were derived.





**Table 6.** Comparison of emission rates derived from emission factors of other studies for two different fuel consumptions. Lower value is for a fuel consumption of $165 \, \mathrm{kg \, h^{-1}}$, which is typical for inland ships. Upper value is for a fuel consumption of $2000 \, \mathrm{kg \, h^{-1}}$ which is roughly the fuel consumption of a large container ship (14.000 TEU carrying capacity) at a speed of $7 \, \mathrm{m \, s^{-1}}$ (Notteboom and Vernimmen, 2009).

| Study | mean $NO_x$ emission factor in $\mathrm{g \, kg^{-1}}$ fuel | $NO_x$ emission rate in $\mathrm{g \, s^{-1}}$ | number of evaluated ships |
|---|---|---|---|
| Moldanová et al. (2009) | 73.4 | 3.4 - 40.8 | 1 |
| Williams et al. (2009) | $66.4 \pm 9.1$ | 3.0 - 36.9 | > 200 |
| Alföldy et al. (2013) | $53.7 \pm 22.3$ | 2.5 - 29.8 | 497 |
| Diesch et al. (2013) | $53 \pm 27$ | 2.4 - 29.4 | 139 |
| Beecken et al. (2014) | $66.6 \pm 23.4$ | 3.1 - 37.0 | 174 |
| Pirjola et al. (2014) | $64.3 \pm 24.6$ | 2.9 - 35.7 | 11 |
| Beecken et al. (2015) | $58 \pm 14.5$ | 2.7 - 32.2 | 466 |
| This study | - | mean 11.0 | 886 |
| | | median 4.6 | |
| | | mean seagoing 10.2 | 632 |
| | | median seagoing 5.2 | |
| | | mean inland 4.5 | 177 |
| | | median inland 1.9 | |

**Table 7.** Estimate of $SO_2$ emission rates for fuels with different fuel sulphur content, calculated for different fuel consumption under the assumption that all sulphur is converted to $SO_2$ during combustion. Lower value is for a fuel consumption of $165 \, \mathrm{kg \, h^{-1}}$, which is typical for inland ships. Upper value is for a fuel consumption of $2000 \, \mathrm{kg \, h^{-1}}$ which is roughly the fuel consumption of a large container ship at a speed of $6 \, \mathrm{m \, s^{-1}}$ (Notteboom and Vernimmen, 2009), which is the typical speed for the largest passing vessels.

| Source | Fuel type | Fuel sulphur content in $\% \, \mathrm{M/M}$ | $SO_2$ emission rate in $\mathrm{g \, s^{-1}}$ |
|---|---|---|---|
| Estimation | SECA limit | 0.1 | 0.09 - 1.16 |
| | Diesel fuel for inland shipping | $1 \times 10^{-5}$ | 0.0009 - 0.0116 |
| This study | - | - | mean 0.44 |
| | | | median 0.25 |
| | | | mean seagoing 0.47 |
| | | | median seagoing 0.28 |
| | | | mean inland 0.10 |
| | | | median inland 0.05 |



The emission rates of inland and seagoing ships have been analysed and compared to each other and showed that sea going ships have higher emission rates than inland ships. Generally the emission rates increase with size and speed of the ship. The uncertainty for a single emission rate are 43 % in the mean and 35 % in median. Repeated measurements of several ships that passed multiple times show a low variability in their emission rates.

To improve the accuracy of the estimate of the ship emission rates, better knowledge of several key parameters will reduce their uncertainty. For example better knowledge of the exact position of the emission location i.e. the position of the ship's funnel, is required. Similarly, better knowledge of the height of the emission, i.e. the height of the funnel of the ship and the water level at the time of measurement, is required. The use of more sophisticated models to describe the shape and evolution of the plume would be of value. Additionally a measurement geometry with a shorter light path across the river would make it

easier to detect the pollution plumes from water craft having small emissions and thereby increase the chances of determining emission rates from such vessels.

In comparison to the standard instrumentation at the measurement site, the LP-DOAS does not need to be calibrated and is able to measure under all wind conditions. However the current LP-DOAS system does not measure $CO_2$, so that relative emission factors cannot easily derived from $NO_x/CO_2$ or $SO_2/CO_2$ ratios. Therefore a model had to be used to calculate the

emission rates of air pollutants. A measurement of the integrated $CO_2$ concentration along the light path would supersede the need for a dispersion model and should be considered for further technical developments of such measurements.

The measurements have demonstrated that accurate emission rates from shipping emissions can be derived from LP-DOAS measurements and that there is much potential in this approach. These emission rates are valuable input for the assessment of the influence of shipping emissions on air quality in regions close to the shipping lanes at the coast or along rivers and canals.

*Author contributions.* KK, SS and FW set up and operated the LP-DOAS instrument. AW provided the data of the in situ measurements. KK performed the analysis of the LP-DOAS data, provided the figures and wrote the manuscript. JB, DP, AR, SS, AW and FW supported the data interpretation. All authors contributed to the writing of the manuscript.

*Competing interests.* The authors declare that they have no conflict of interest.

*Acknowledgements.* The research project which facilitated the reported study was funded in part by the German Federal Maritime and Hy-
drographic Agency (Bundesamt für Seeschifffahrt und Hydrographie, BSH) and the University of Bremen. The authors thank the Waterways and Shipping Office Hamburg for their help and support.





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
