# Peer review of "Estimation of ship emission rates at a major shipping lane by long path DOAS measurements"

_Atmospheric Measurement Techniques, 2021_

## Referee Comment (RC1)

The authors reported a new program for studying ship emission rates based on active remote sensing observations. Monitoring of ship emissions is usually carried out using in situ instruments on land, which depend on favourable wind conditions to transport the emitted substances to the measurement site. LP-DOAS measurements overcomes this shortcoming, and it realized real-time observation. However, there are some issues that need to be explained by the authors. After these minor corrections, this manuscript can be accepted by AMT.

1. The measured $NO_x$ and $SO_2$ concentration and their emission rates should be validated with the measured data.

2. How to characterize the emission concentration and emission rates of $NO_x$ and $SO_2$ at the ship chimney mouth?

3. During the analysis, the authors should quantify the impact of $NO_x$ photolysis.

---

## Author Comment (AC1)

We would like to thank the reviewer for his / her useful comments.

**1. The measured NOx and SO2 concentration and their emission rates should be validated with the measured data.**

Comparison of the concentrations of the in situ measurements and the DOAS measurements is not straight forward, as both systems measure different air masses. The insitu measurements rely on the transport of air masses to the measurement site, while the LP-DOAS measures the integrated number density of the respective absorbers along the light path. As only a small portion of the light path is affected by a plume, the LP-DOAS measures a lot of background concentration and possible enhancements of $NO_2$ or $SO_2$ are lower than for the insitu instruments. Nevertheless both measurement systems show similar results (e.g. Figure 1).

Modified Section 2.4 to include this information.

**2. How to characterize the emission concentration and emission rates of NOx and SO2 at the ship chimney mouth?**

The presented emission rates are the emission rates encountered at the ships chimney mouth.

The emission concentration at the ships chimney mouth depend on the dimensions of the ships chimney. Comparing two chimneys with different sizes, but same emission rate would lead to different emission concentrations at the mouth of the chimney, where the smaller chimney would have higher concentrations compared to the larger one, but their mass flow through the cross section of the chimney mouth would be the same.

**3. During the analysis, the authors should quantify the impact of $NO_x$ photolysis.**

Generally the plumes are measured quite shortly after their emission. The time difference between measured peak and assigned AIS position is between 4 to 100 seconds with a median and mean of about 20 seconds. Therefore the effects of $NO_2$ photolysis are small compared to other influencing factors and can be neglected.

Modified Section 2.4.3 to include this information.

[Figure]

**Figure 1:** Time series of $NO_2$, $O_3$ and $SO_2$ measured by the insitu instruments (blue) and the LP-DOAS (orange) on 20th July 2018.

---

## Author Comment (AC2)

We would like to thank the reviewer for his / her useful comments.

**Sect. 2.4.: Please describe generally the spectral analysis performance, e.g. how about the residual and the fit errors for each species? And any filtering applied for measured data before introducing the inversion program.**

Typical DOAS measurement errors, which are defined as two times the DOAS fit error, can be found in Table 5, and typical detection limits are mentioned in section 2.4.

Added the following to Sect. 2.4:

"Before further analysis of the trace gas time series, fits with an RMS higher than 0.01 are removed. These high RMS usually occur when the ship blocks the light path."

**Sect. 2.4.1: If there were averagely 110 ship passages per day and 200 days measured data were analyzed, does it mean that only 30 % success rate of the identification, e.g. 7402/(110*233). I think the authors could discuss more details about this or any explanations, which may be related the performance of the identification algorithm.**

The algorithm tries to identify ship positions close to the light path in a given time window around a peak occurrence in the trace gas time series. Within this window a ship is only assigned to a peak if there is no other ship nearby. The final position is the first AIS position, where the distance to the light path is equal to or longer than the ships length. As the position of the AIS receiver in relation to the ships chimney is not known, this approach ensures that the ships chimney also passed the light path and that the modelled plume of the Gaussian Plume Model passes through the light path. Neglecting the additional criterion of a full pass and using stricter time windows around each peak, a higher number of peaks could be attributed to ships, but this also increases the chance of mismatches and the assignment of mixed plumes of several ships to a single ship.

Added the following sentence to Section 2.4.1:

"Neglecting the additional criterion of a full pass and using stricter time windows around each peak, a higher number of peaks could be attributed to ships, but this also increases the chance of mismatches and the assignment of mixed plumes of several ships to a single ship."

**Sect. 2.4.2: If I do understand correctly, the authors used NO2/NOx ratio is provide by the in-situ measurement at river side, which is the aged plume rather than the fresh plume at the chimney. The difference of NO2/NOx ratio between fresh and aged plume will result in the larger uncertainties on the conversion of NO2 to NOx. In addition, the authors need to check the dependence of in-situ measured NO2/NOx on the ship position and wind direction.**

There seems to be no indication of dependence of the $NO_2/NO_x$ ratio on the ships position, wind direction or age of the plume (see Figures 1 to 4). However due to the reaction of NO with $O_3$ that forms $NO_2$, the decrease in $O_3$ has to be considered, as mentioned in the manuscript (see Figure 5).

Added the following sentence to section 2.4.2:

"There is no indication for further dependencies of the $NO_2/NO_x$ ratio on the position of the source ship, the wind direction or the age of the plume."

**Sect. 2.4.3: Any introduction for Equation 6 and relevant parameters? Moreover, considering the movements of ships and continuous emission of chimney, the detected plume by LP-DOAS at given time is not only the pure emission of the start point, but also mixed with the subsequent ship plume during the cruise. Did the authors consider this condition in the Gaussian plume model estimation? If not, at least the authors should take an example to evaluate the effects on the model evaluation.**

Equation 6 calculates the direction the apparent wind at the ships position, $v_{windE}$, $v_{shipE}$ and $v_{windN}$, $v_{shipN}$ are the eastern and northern velocity components of the wind vector and ship movement vector, respectively. The atan2 expression is a generally available variation of the arctangent function which returns the inverse tangent of the first and second argument to the function (Berg et al., 2012).

Yes, the detected plume measured by the LP-DOAS is not only the pure emission of the start point, but also subsequent emissions during the cruise and this should be considered in the model. Depending on the course of the ship two cases can be distinguished. In the first case the plume is more or less orthogonal to the light path and the measured parts of the plume have more or less the same age (see Figure 6). In the second case the plume is not orthogonal to the light path and newer parts and older parts of the plume get measured at the same time

[Figure]

**Figure 1:** $NO_2/NO_x$ ratio in dependence of the distance between ship and measurement site.

(See Figure 7). The modelled region covers an area of approximately 2800 m x 700 m and the assigned ship position is always very close to the light path. Therefore the slightly different time of emission is neglected for simplicity and it is assumed that the measured plume is the result of the pure emission at the start point.

Section 2.4.3 has been modified to include this information.

**Table 2, please specify the temperature of the used absorption cross section.**

Changed the respective table to include this information.

[Figure]

**Figure 2:** $NO_2/NO_x$ ratio in dependence of the age of the plume (travel time from source to measurement site).

[Figure]

**Figure 3:** $NO_2/NO_x$ ratio in dependence on wind direction.

[Figure]

**Figure 4:** Scatterplot of ship positions used for derivation of the $NO_2/NO_x$ ratios. $NO_2/NO_x$ ratios are colour coded.

[Figure]

**Figure 5:** Plot of $\Delta NO_2 + \Delta O_3$ against $\Delta NO_x$ from peaks measured with the in situ instruments between April 2018 and May 2019. All concentrations have been corrected for background concentrations. For this analysis, 220 manually quality checked peaks were used. This results in a slope (a $NO_2/NO_x$ ratio) of 0.138 with a respective standard error of 0.006.

[Figure]

**Figure 6:** Example case for a plume being orthogonal to the light path. Upper left panel shows the route the ship is taking through the light path. Upper right panel shows the plume simulation used to derive the emission rate. Lower panel shows the measured ship peak. The measured parts of the plume have similar age.

[Figure]

**Figure 7:** Example case for a plume being non-orthogonal to the light path. Upper left panel shows the route the ship is taking through the light path. Upper right panel shows the plume simulation used to derive the emission rate. Lower panel shows the measured ship peak. Different parts of the plume, with different age, are measured at the same time.

**References**

Berg, N., Mellqvist, J., Jalkanen, J.-P., and Balzani, J.: Ship emissions of SO 2 and NO 2 : DOAS measurements from airborne platforms, Atmospheric Measurement Techniques, 5, 1085–1098, https://doi.org/10.5194/amt-5-1085-2012, 2012.

---

## Author Response (AR2)

Dear Dr. Folkert Boersma,

Thank you for accepting our manuscript for publication in AMT. The following corrections and clarifications have been made for the final manuscript.

Yours sincerely,
Kai Krause

**L98: I don't think the "insitu" (please write in situ) measurements and site have been introduced at this stage yet.**
Changed to in situ and verified that the in situ measurements have already been mentioned previously (chapter 2.1 Measurement site).

**L122: windows --> window**
Changed.

**2.4.4 Title: NO2 to NOx conversion is unfortunate, since NO2 is an intrinsic part of NOx by definition. Please consider rephrasing.**
Changed to "Estimation of $NO_x$ from measured $NO_2$".

**L178: suggest to write atan2 not in italic font.**
Changed.

**L226: furthermore is one word, as is underrepresented in L305.**
Changed.